# Overcoming the low reactivity of biobased, secondary diols in polyester synthesis

**Daniel H. Weinland** [1], **Kevin van der Maas**[1], **Yue Wang** [1], **Bruno Bottega Pergher** [1], **Robert-Jan van Putten**[1,2], **Bing Wang**[2] & **Gert-Jan M. Gruter** [1,2] ✉

Shifting away from fossil- to biobased feedstocks is an important step towards a more sustainable materials sector. Isosorbide is a rigid, glucose-derived secondary diol, which has been shown to impart favourable material properties, but its low reactivity has hampered its use in polyester synthesis. Here we report a simple, yet innovative, synthesis strategy to overcome the inherently low reactivity of secondary diols in polyester synthesis. It enables the synthesis of fully biobased polyesters from secondary diols, such as poly(isosorbide succinate), with very high molecular weights ($M_n$ up to 42.8 kg/mol). The addition of an aryl alcohol to diol and diacid monomers was found to lead to the in-situ formation of reactive aryl esters during esterification, which facilitated chain growth during polycondensation to obtain high molecular weight polyesters. This synthesis method is broadly applicable for aliphatic polyesters based on isosorbide and isomannide and could be an important step towards the more general commercial adaption of fully biobased, rigid polyesters.

Polymer materials are an indispensable part of modern life owing to their wide range of tuneable properties. The materials sector currently uses mainly fossil resources (-99%) as a feedstock for polymer synthesis, although significant efforts are being undertaken to switch to more sustainable feedstocks[1]. A shift away from fossil and towards biomass leads to challenges, but also opportunities: New types of monomers become available that potentially lead to materials with improved properties[2]. Condensation polymers like polyesters are especially interesting due to their high oxygen content, which ties into the high level of oxygenation of bioderived monomers[3], and inherent recyclability[4]. In terms of improving mechanical-, barrier-, and thermal properties, the glucose-derived rigid isosorbide is a highly promising polyester monomer[5,6]. It is already produced on a 25 kiloton scale, but the problem is that due to it being a secondary diol, its reactivity is very low, which in turn limits its application to a minor component in copolyesters[7].

For the past three decades, scientists have been unsuccessfully trying to synthesize fully biobased poly(isosorbide succinate) (PIsSu). Thus far, no molecular weights high enough for thermoplastic applications have been reported due to the low reactivity of isosorbide in

(trans)esterifications, even with highly reactive diacid dichlorides (Fig. 1a). Applications of PIsSu have been limited to powder coatings[8], which do not require a high molecular weight polymer[9].

Regardless of the diacid comonomer, there have been no reports of fully isosorbide-based polyesters with a $M_n > 23.0$ kg/mol, despite the first reported synthesis of isosorbide in the 1950's and a number of efforts to overcome the low reactivity of isosorbide with unconventional synthesis strategies (Fig. 1b). These low molecular weights lead to brittle polymer products, which cannot be used in most thermoplastic applications. This can also be observed in the large variation of $T_g$ values reported for PIsSu (Fig. 1a), which is an indication that a plateau value of the molecular weight has not yet been obtained to reach the "true" $T_g$ value of PIsSu. Current applications of isosorbide in polyester synthesis are thus limited to copolyesters with low molar percentages (<40 mol%) of isosorbide relative to total diol. Even though the incorporation of isosorbide into polyesters has been shown to improve their thermal and mechanical properties, up to now the properties of fully isosorbide-based polyesters are largely unexplored.

In the present work, an attempt was made to overcome the low reactivity of isosorbide relative to water by using aryl alcohols as

[1]Van't Hoff Institute of Molecular Sciences, University of Amsterdam, P.O. Box 94720, 1090GS Amsterdam, The Netherlands. [2]Avantium Chemicals BV, Zekeringstraat 29, 1014BV Amsterdam, The Netherlands. ✉e-mail: g.j.m.gruter@uva.nl

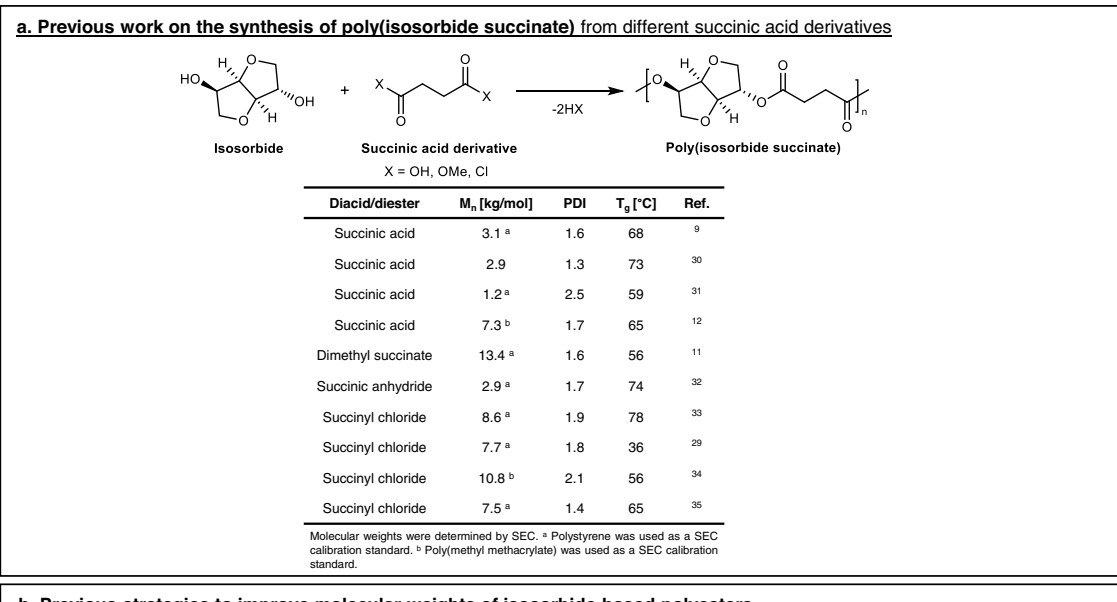

Molecular weights were determined by SEC. a Polystyrene was used as a SEC calibration standard. b Poly(methyl methacrylate) was used as a SEC calibration standard.

**Fig. 1 | Previous literature and this work. a** Comparison of previous literature on the synthesis of poly(isosorbide succinate) (PIsSu)[9,11,12,29–35]. **b** Previous strategies to improve molecular weights of isosorbide-based polyesters[16,24,36]. **c** The synthesis strategy of this work. Not shown in **b** is a strategy reported by Kricheldorf et al.[37]. on the improvement of isosorbide-based polyester molecular weights via silylation of isosorbide's-OH groups and subsequent reaction with diacid chlorides. Please note that only one regioisomeric combination of isosorbide repeat units is depicted in the polymer structures. The chirality of isosorbide will result in a polymer with a number of possible arrangements that do not necessarily possess tacticity.

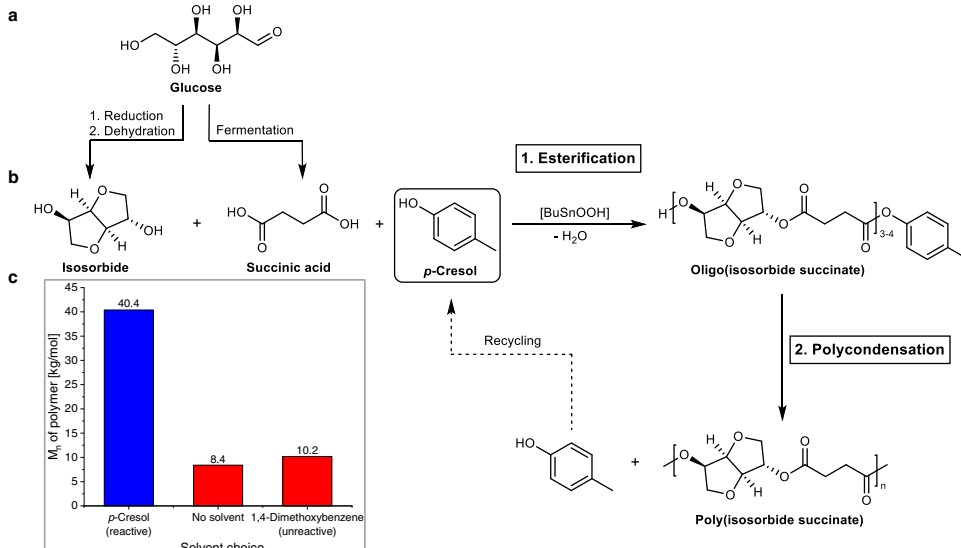

**Fig. 2 | Synthesis of high molecular weight PIsSu with aryl alcohol. a** Monomer synthesis from glucose, obtainable from either first or second-generation biomass. **b** Synthesis of PIsSu from succinic acid, isosorbide and *p*-cresol (see Fig. 3 for more information on the esterification). **c** PIsSu molecular weights obtained with a reactive solvent (*p*-cresol), no solvent, and an unreactive solvent (1,4-dimethoxybenzene) under comparable reaction conditions.

reactive solvents to simultaneously remove water and form reactive aryl ester end groups. The reaction was performed in glassware and also upscaled to a 2 L autoclave. PIsSu was the main target material, but also other polyesters were successfully synthesized using both another 1,4:3,6-anhydrohexitol (isomannide) and several aliphatic diacids. The barrier and mechanical properties of a number of polyesters synthesised with this strategy were studied.

## Results

### Effect of aryl alcohol on molecular weight

Figure 2 shows the reaction strategy for improving the molecular weights of polyesters based on isosorbide. The addition of *para*-methyl phenol (*p*-cresol) to isosorbide and succinic acid under esterification reaction conditions was found to lead to the formation of oligo(isosorbide succinate) units with *p*-cresyl succinate ester end groups (Fig. 2b). This was expected, as aryl alcohols are typically considered chain-stopping moieties in step growth polymers due to their monofunctionality[10]. During pre-polycondensation (400–6.5 mbar), unreacted *p*-cresol was removed from the reaction melt and within one hour at full vacuum (<1 mbar), a highly viscous polymer melt was obtained. Analysis of the final product revealed a number average molecular weight of 40.4 kg/mol. This is a very high value for a polyester containing 100 mol% of isosorbide as the diol moiety (highest reported $M_n$ for PIsSu: 13.4 kg/mol[11]). The polyester had a $T_g$ of 82 °C, which is higher than previously reported $T_g$ values for PIsSu (Fig. 1a, previously reported $T_g$ values of PIsSu reach as low as 36 °C and never higher than 73 °C). Typically publications on fully isosorbide-based polyesters synthesized from carboxylic acids or carboxylic acid alkyl esters under standard conditions report polycondensation times between 4 and 8 h, reaching $M_n$ values not higher than 20 kg/mol[7,11–13].

Aryl esters are known to be more reactive in transesterification reactions than alkyl esters or carboxylic acids due to the improved leaving group ability of aryl alcohols and the unfavourable back reaction after transesterification with water or an aliphatic alcohol[14]. Their use in polyester synthesis, in the form of diphenyl terephthalate, has been described in patents from the 1960s by Eastman, in order to improve molecular weights of polyesters based on the unreactive secondary diol 2,2,4,4-tetramethyl-1,3-cyclobutanediol[15]. Some recent patents describe the use of diphenyl oxalate for the synthesis of polyesters based on thermally labile oxalic acid, isosorbide, and a

linear diol comonomer[16] (Fig. 1b). Traditionally, aryl esters are synthesized from carboxylic acid chlorides[14] or carboxylic acids[17] in a separate reaction step and require separate purification steps prior to polyester synthesis, as well as the use of stoichiometric reagents.

### Effect of aryl alcohols during esterification and polycondensation

Subsequently, more in-depth research on this synthesis strategy was performed. After 5 h of esterification between isosorbide, succinic acid, and *p*-cresol (1:1:1.5 molar ratio) at 240 °C a steady state was reached: $H_2O$ was removed from the reaction mixture as condensation product and oligo(isosorbide succinate) units with a number average molecular weight of around 1.6 kg/mol were obtained. Structure elucidation of the reaction mixture by ¹H NMR revealed the complete conversion of succinic acid's carboxylic acid end groups to *p*-cresyl succinate groups (Fig. 3a, b). Neither the oligomer length nor the alcohol-to-ester end group ratio changed significantly with longer reaction times (Fig. 3c, d).

The addition of such significant amounts of aryl alcohols to the monomers during esterification can potentially have two effects: (1) Reduction of melt viscosity, accompanying improved mass transfer and improved removal of $H_2O$, thus driving the esterification equilibrium forward. (2) Formation of the observed *p*-cresyl succinate end groups with a superior reactivity compared to carboxylic acid end groups, which at the same time eliminates water formation and the accompanying backward reactions during polycondensation, as water is much more reactive than aryl alcohol.

Some control experiments were conducted to find out what contributes to the high molecular weights that were obtained. Reactions without any solvent yielded a low molecular weight product ($M_n$ = 8.4 kg/mol, Fig. 2c), which is in line with previous work on PIsSu. To rule out (1), a reaction with the non-reactive solvent 1,4-dimethoxybenzene was conducted, which has a boiling point similar to *p*-cresol. Oligomers with a $M_n$ of 5.0 kg/mol were obtained after 5 h esterification, which is significantly higher than in reactions with the reactive solvent *p*-cresol ($M_n$ = 1.6 kg/mol). This is likely due to the non-reactive nature of 1,4-dimethoxybenzene, which does not inhibit the chain growth of oligomers during esterification. The final product however had a significantly lower molecular weight ($M_n$ = 10.2 kg/mol) compared to a reaction conducted with *p*-cresol (Fig. 2c). These

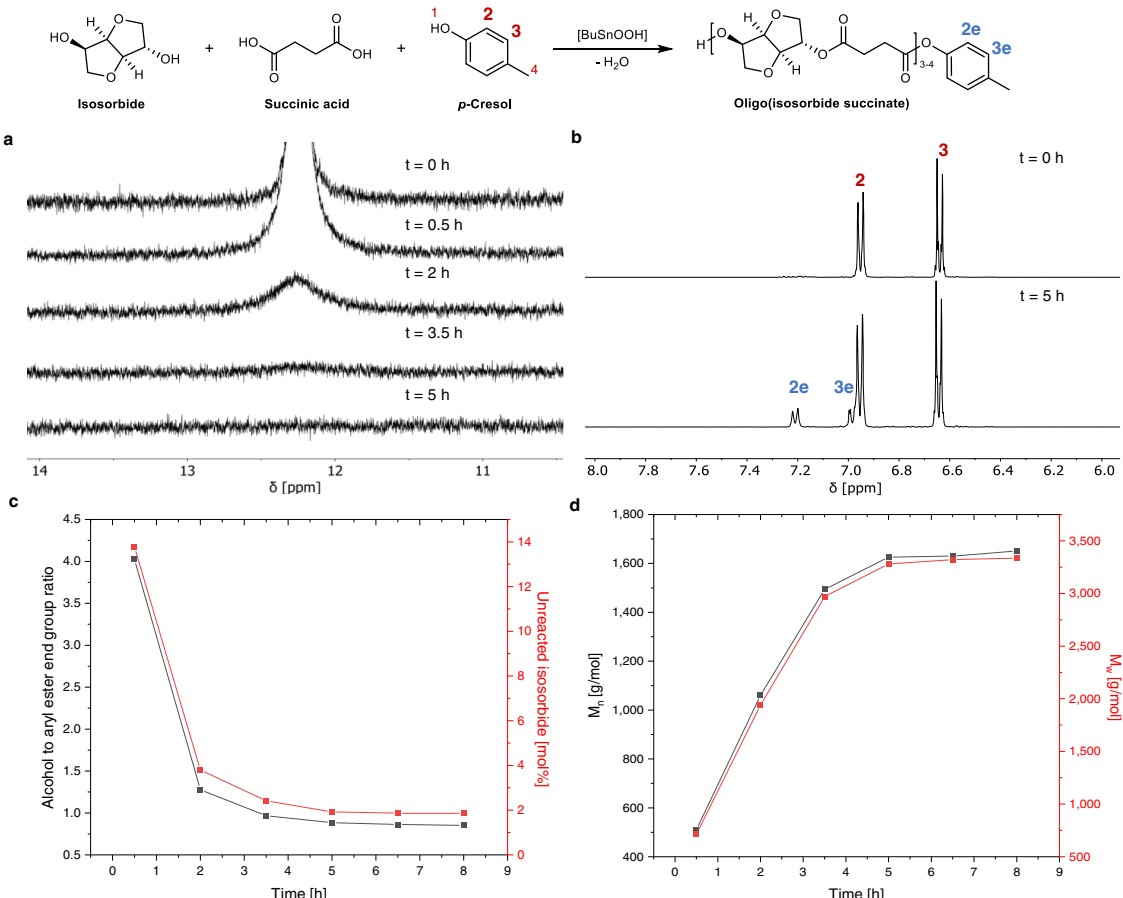

**Fig. 3 | Esterification between succinic acid, isosorbide, and *p*-cresol. a** ¹H NMR spectrum showing the disappearance of succinic acid's carboxylic acid proton around 12 ppm after 0 h, 0.5 h, 2 h, 3.5 h, and 5 h. **b** ¹H NMR spectrum of aromatic region upon formation of a clear melt (*t* = 0) and after 5 h esterification at 240 °C but before *p*-cresol removal. For a fully assigned ¹H NMR spectrum of the reaction melt after esterification, see Supplementary Fig. 2. **c** Evolution of the alcohol (*exo*- and *endo*-OH groups of isosorbide) to ester end group (cresyl succinate) ratio during esterification with the amount of total unreacted isosorbide (respective

total mol% of isosorbide). Determined from the ¹H NMR spectra of the reaction melt taken after different esterification times (see Supplementary Fig. 1). For an assignment of the different end groups and signals corresponding to unreacted isosorbide used for the calculations, see Supplementary Fig. 2. For equations used to calculate end group ratios and unreacted isosorbide, see Supporting Information p.5. **d** Molecular weight evolution of oligo(isosorbide succinate) during esterification determined by GPC. Both **c** and **d** show that a steady state is reached after ~5 h of esterification.

results, together with the known high reactivity of aryl esters in transesterification reactions[14], confirm the importance of *p*-cresyl succinate ester groups for a successful polymerization.

It should be noted that an equimolar ratio of isosorbide and succinic acid is essential to obtain a high molecular weight polymer product. Using an excess of isosorbide, as is commonly reported in synthesis procedures of isosorbide-based polyesters, leads to an excess of alcohol end groups during polycondensation. This in turn reduces the molecular weights that can be obtained during the last stages of the reaction, as excess isosorbide must be replaced in order to grow the polymer chain. This is exceedingly difficult with isosorbide due its high boiling point and low reactivity. Using an excess of isosorbide thus effectively negates the benefits of in situ generated aryl ester end groups in the present synthesis strategy.

One important factor when applying the principles of Circular Chemistry[18] to this synthesis strategy is the recyclability of the aryl alcohol. During esterification between diacid, diol, and aryl alcohol, water is formed as a condensation product, which is removed from the reactor and collected in the receiving flask together with small amounts of aryl alcohol. A biphasic system is formed, as water and *p*-cresol are largely immiscible at room temperature. During pre-polycondensation and polycondensation, the remaining aryl alcohol is collected in the receiving flask. Recycling experiments of the aryl

alcohol were conducted without separation of the water layer prior to subsequent reactions. Any additional water from previous experiments added at the beginning of the reaction is rapidly removed together with water formed during esterification. If desired, both fractions can be easily separated by either removing water from the receiving flask prior to polycondensation or phase separation after polymerization. *p*-Cresol can be recovered unaltered in both appearance and the chemical composition after polycondensation (Fig. 4b). A negligible amount of succinic anhydride was identified in *p*-cresol recovered after polycondensation (Fig. 4c). To confirm its reusability, a set of recycling experiments was conducted without any additional purification (Fig. 4a). Reusing *p*-cresol up to five times did not result in a significantly lower molecular weight product.

## Optimization of reaction conditions
The reaction conditions with regard to the type and amount of aryl alcohol, reaction temperature, and catalyst choice were optimized next. Initial experiments (not shown) revealed that high esterification temperatures of 240 °C were required to achieve full conversion of succinic acid -COOH groups to aryl ester end groups within reasonable reaction times. When choosing the type of aryl alcohol to optimize the reaction conditions, several factors were considered: (I) The aryl alcohol should have a sufficiently high ambient pressure boiling point

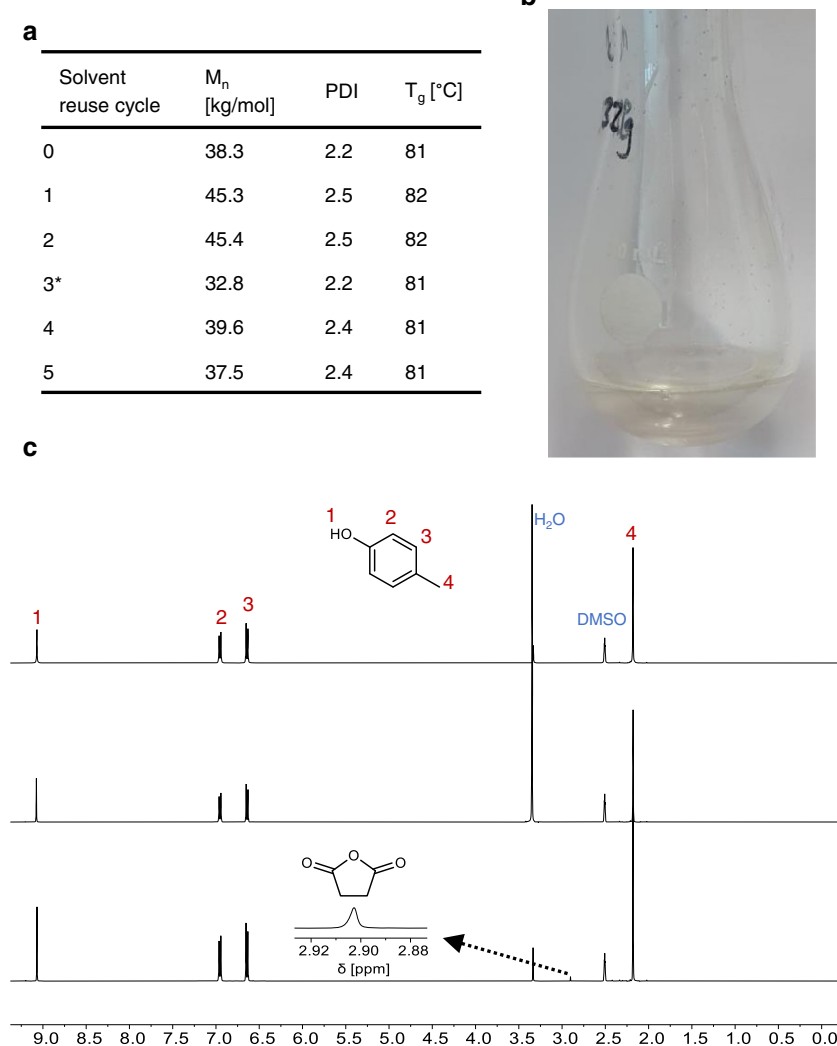

| a | | | |
|---|---|---|---|
| Solvent reuse cycle | $M_n$ [kg/mol] | PDI | $T_g$ [°C] |
| 0 | 38.3 | 2.2 | 81 |
| 1 | 45.3 | 2.5 | 82 |
| 2 | 45.4 | 2.5 | 82 |
| 3* | 32.8 | 2.2 | 81 |
| 4 | 39.6 | 2.4 | 81 |
| 5 | 37.5 | 2.4 | 81 |

**Fig. 4 | Reuseability of *p*-cresol. a** Molecular weight of PIsSu synthesized with recycled *p*-cresol. *p*-Cresol was collected after polycondensation and reused without further purification. After five reuse cycles of *p*-cresol, no significant change in the molecular weight of the final product was observed. Differences in molecular weight can arise due to the very high melt viscosity of PIsSu, which can prevent stirring in the last 30 min of high vacuum polycondensation. *A leak occurred in the reactor setup during high vacuum, which resulted in a lower molecular weight. **b** Picture of *p*-cresol after 5 reuse cycles. No change in appearance was visible to *p*-cresol from the bottle except for a water layer that formed during esterification. Please note that recovered *p*-cresol was stored under nitrogen in the dark to prevent discolouration. **c** ¹H NMR spectra of fresh *p*-cresol (top), *p*-cresol recovered after pre-polycondensation (>6.5 mbar, middle), and *p*-cresol recovered after high vacuum polycondensation (<1 mbar, bottom). The ¹H NMR spectra of recovered *p*-cresol were measured after five reuse cycles. ¹H NMR analysis indicated very small amounts of succinic anhydride, which was collected during high vacuum polycondensation (<1 mbar).

to enable esterification at 240 °C. This (and (II)) precluded the use of unsubstituted phenol, which rapidly boils out of the reactor at 240 °C and does thus not correspond to the reaction parameters required for successful esterification. (II) In accordance with the principles of Green Chemistry, the aryl alcohol should not be a CMR compound, exhibit acute toxicity or be bioaccumulative. This excluded the use of halogenated phenols. (III) The aryl alcohols should be readily commercially available, as the goal of this work was to establish an industrially feasible synthesis strategy. This precluded the use of *meta*-substituted derivatives. With these limitations in mind, we set forth to investigate two additional aryl alcohols which fit these requirements (Table 1).

Despite full conversion of -COOH groups during esterification in a reaction conducted with *p*-methoxyphenol, its high boiling point hampered its removal during polycondensation, which resulted in a lower molecular weight polymer (28.9 kg/mol vs. 40.4 kg/mol with *p*-cresol). The use of *o*-methoxyphenol (guaiacol) resulted in incomplete conversion of -COOH groups after 5 h esterification, likely caused by the larger steric hindrance compared to *para*-substituted aryl alcohols. Subsequent polycondensation yielded a lower molecular weight product (16.9 kg/mol) than reactions with *para*-substituted phenol derivatives. This is likely related to the incomplete conversion of -COOH groups to aryl ester groups after esterification, which hinders chain growth during polycondensation. No other aryl alcohols were investigated as they did not fit the requirements stated above.

The amount of *p*-cresol required for a successful reaction was found to be 1.5 equiv. respective to the diacid moiety (Table 2). It was observed that the amount of solvent also had an effect on the amount of unreacted isosorbide after esterification. This is due to the alcohol functionality of aryl alcohols, which leads to esterification with succinate units, which in turn can replace isosorbide.

A pre-polycondensation and polycondensation temperature of 240 °C resulted in a lower molecular weight polymer ($M_n$ = 29.9 kg/mol vs. $M_n$ = 40.4 kg/mol at 220 °C). This is likely related to the thermal

**Table 1 | Influence of aryl alcohol on poly(isosorbide succinate) molecular weight and $T_g$**

| Solvent | Boiling point [°C] | $M_n$ [kg/mol] | PDI | $T_g$ [°C] |
|---|---|---|---|---|
| / | / | 8.4 | 2.4 | 76 |
| p-Cresol | 202 | 40.4 | 2.4 | 82 |
| p-Methoxyphenol | 243 | 28.9 | 2.1 | 82 |
| o-Methoxyphenol | 205 | 16.9 | 2.0 | 80 |
| 1,4-Dimethoxybenzene (unreactive) | 213 | 10.2 | 2.0 | 78 |
| Diphenyl ether (unreactive) | 258 | 10.2 | 2.0 | 78 |

Reaction conditions (unless noted otherwise): Reactions were conducted on a 60 mmol scale (respective diacid), with a 1:1:1.5 molar ratio of isosorbide:succinic acid:solvent and 0.1 mol% BuSnOOH catalyst (resp. diacid). Esterification was conducted for 5 h at 240 °C. For pre-poly-condensation, the temperature was decreased to 220 °C and the pressure reduced from atmospheric pressure to 0.4–0.8 mbar within 2 h. Polycondensation was conducted for 1 h at 220 °C and 0.4–0.8 mbar.

**Table 3 | Influence of catalyst on molecular weight and $T_g$ of PIsSu**

| Catalyst | n (catalyst) [mol%]* | ppm (metal) | $M_n$ [kg/mol] | PDI | $T_g$ [°C] |
|---|---|---|---|---|---|
| / | / | / | 3.7 | 2.1 | 70 |
| BuSnOOH | 0.1 | 520 | 40.4 | 2.4 | 82 |
| BuSnOOH | 0.05 | 260 | 34.7 | 2.2 | 82 |
| Ti(OBu)₄ᵃ | 0.1 | 210 | 33.0 | 2.2 | 83 |
| Ti(OBu)₄ᵃ | 0.05 | 105 | 28.7 | 2.1 | 81 |
| Zr(OBu)₄ | 0.1 | 400 | 3.7 | 2.1 | 79 |
| Zn(OAc)₂ | 0.1 | 287 | 1.7 | 2.1 | 72 |
| GeO₂ | 0.1 | 318 | 1.5 | 2.0 | 59 |

Reaction conditions (unless noted otherwise): Reactions were conducted on a 60 mmol scale (respective diacid), with a 1:1:1.5 molar ratio of isosorbide:succinic acid:p-cresol and the indi-cated catalyst. Esterification was conducted for 5 h at 240 °C. For pre-polycondensation, the temperature was decreased to 220 °C and the pressure reduced from atmospheric pressure to 0.4–0.8 mbar within 2 h. Polycondensation was conducted for 1 h at 220 °C and 0.4–0.8 mbar.
*Mol% respective diacid.
ᵃCatalyst was added in two portions to compensate for hydrolytic sensitivity of Ti-alkoxides.

**Table 2 | Influence of p-cresol equivalents on molecular weight and $T_g$ of PIsSu**

| Equiv. p-cresol | $M_n$ [kg/mol] | PDI | $T_g$ [°C] | Unreacted isosorbide after esterification [mol%]* |
|---|---|---|---|---|
| 0.5 | 28.1 | 2.5 | 81 | 0.8 |
| 0.9 | 40.3 | 2.7 | 83 | 1.2 |
| 1.5 | 40.4 | 2.4 | 82 | 2.0 |
| 2.2 | 31.6 | 2.0 | 82 | 2.9 |

Reaction conditions (unless noted otherwise): Reactions were conducted on a 60 mmol scale (respective diacid), with a 1:1 molar ratio of isosorbide:succinic acid and 0.1 mol% BuSnOOH catalyst (resp. diacid). The amount of p-cresol in each experiment was varied as indicated (respective diacid). Esterification was conducted for 5 h at 240 °C. For pre-polycondensation, the temperature was decreased to 220 °C and the pressure reduced from 0.4–0.8 mbar within 2 h. Polycondensation was conducted for 1 h at 220 °C and 0.4–0.8 mbar.
*Determined from 1H NMR spectra after esterification, see Supplementary Information p. 5.

lability of succinic acid-based polyesters under high vacuum conditions. Previous work on poly(1,4-butylene succinate) revealed that succinic anhydride is liberated during polycondensation at 240–260 °C[19]. Similarly, we found that succinic anhydride was liberated from the polymer during polycondensation, which is facilitated by elevated temperatures. This thermal lability of the succinate repeat unit further underlines the importance of p-cresyl esters formed during esterification, as they do not require high reaction temperatures to react with isosorbide's unreactive secondary alcohol groups.

Lastly, the influence of the catalyst choice on the reaction outcome was investigated (Table 3). A reaction conducted without a catalyst resulted in incomplete conversion of -COOH groups to aryl ester groups, and a low molecular weight final product (3.7 kg/mol). Butyltin hydroxide oxide (BuSnOOH) yielded the highest molecular weights. The more benign Ti(OBu)₄ yielded polymers with a slightly decreased molecular weight (33.0 kg/mol). Upon addition of the catalyst to the reaction mixture, a deep red colouration was observed. This colour persisted in the final polymer product and is likely caused by the formation of titanium(IV) tetra(p-cresolate). Other catalysts (Zr(OBu)₄, Zn(OAc)₂, and GeO₂) yielded very low molecular weight polymers.

The influence of other reaction parameters like monomer purity and addition of thermal stabilizers was investigated next (Table 4). A higher monomer purity and conducting the reaction under exclusion of oxygen led to a slight increase in the molecular weight of the final product (Table 4, entry 2). Addition of Irgafos 168 to reactions with p-cresol improved the colouration of PIsSu (Table 4, entry 3).The molecular weight of PIsSu in reactions with the more benign guaiacol could

be improved to 32.6 kg/mol with an adjustment of reaction conditions (Table 4, entry 5). The final product did, however, show a dark brown colouration when compared to reactions with p-cresol. Strong discolouration occurred in all butyltin hydroxide oxide catalysed reactions after 5 h esterification with guaiacol, regardless of monomer purity and addition of thermal stabilizers. It should be noted that PIsSu in general should be colourless, the presence of small quantities of coloured by-products causes an orange-yellow colour. This is not unusual in early stage research on (isosorbide-based) polyesters, and it can typically be improved through further research. The use of guaiacol is more desirable in future applications as it is more benign than p-cresol and it can be synthesized from lignocellulosic biomass. Due to shorter reaction times and higher molecular weight products, p-cresol was used to explore the scope of this method and to conduct initial upscaling experiments.

**Monomer scope**

The scope of the p-cresol assisted polyester synthesis with isosorbide and isomannide was extended to other relevant diacids (Table 5).

The results in Table 5 clearly show that the described method can be applied to a broad range of aliphatic diacids. To the best of our knowledge some polyesters are reported here for the first time. For the polyesters that have been previously described in the literature, the molecular weights from the present study are generally much higher than those from the earlier reports. This is despite the fact that most compositions reported previously have only been synthesized from their respective diacid dichlorides, which have a very high reactivity. It is furthermore important to note that polyester synthesis from diacid dichlorides is not economically viable on a commercial scale. The reaction conditions generally had to be adjusted for the different monomers (see Supplementary Table 1 for more details). Esterification was typically conducted until a steady state was reached (see Supplementary Figs. 3 and 4 for a comparison of ¹H NMR spectra taken after esterification for polyesters described in Table 5). All reported compositions yielded ductile polymer products, except the new polyester poly(isosorbide thiodiglycolate), which could not be synthesized with sufficiently high molecular weights due to the low thermal stability of thiodiglycolic acid.

**Barrier- and mechanical properties of isosorbide-based polyesters**

After successful optimization of the reaction conditions, the material properties of isosorbide-based polyesters were investigated to find potential applications (Fig. 5).

**Table 4 | Influence of monomer purity, thermal stabilizers, and aryl alcohol choice on poly(isosorbide succinate) molecular weight**

| Entry | Monomer purity | Solvent | Additive | ppm (additive) | $M_n$ [kg/mol] | PDI | $T_g$ [°C] |
|---|---|---|---|---|---|---|---|
| 1 | Regular[a] | p-Cresol | / | / | 40.4 | 2.4 | 82 |
| 2 | High[b] | p-Cresol | / | / | 42.8 | 2.0 | 82 |
| 3 | High[b] | p-Cresol | Irgafos 168 | 496 | 38.9 | 2.0 | 81 |
| 4 | High[b] | p-Cresol | Irganox 1010 | 499 | 38.7 | 2.0 | 82 |
| 5 | High[b] | Guaiacol[c] | Irgafos 168 | 496 | 32.6 | 1.9 | 82 |
| 6 | High[b] | Guaiacol[c] | Irganox 1010 | 499 | 32.4 | 2.0 | 82 |
| 7 | High[b] | Guaiacol[c] | /[d] | / | 7.1 | 2.4 | 78 |

Reaction conditions (unless noted otherwise): Reactions were conducted on a 60 mmol scale (respective diacid), with a 1:1:1.5 molar ratio of isosorbide:succinic acid:p-cresol and 0.1 mol% BuSnOOH catalyst (resp. diacid). Esterification was conducted for 5 h at 240 °C. For pre-polycondensation, the temperature was decreased to 220 °C and the pressure reduced from atmospheric pressure to 0.4–0.8 mbar within 2 h. Polycondensation was conducted for 1 h at 220 °C and 0.4–0.8 mbar.

[a]Regular succinic acid and in-house purified isosorbide (see Materials) were used.

[b]High purity monomers were used (see Materials) and the reaction was degassed at 80 °C (4 vacuum/nitrogen cycles) before starting esterification.

[c]Esterification was conducted for 9 h instead of 5 h, polycondensation was conducted for 1.5 h instead of 1 h.

[d]Esterification was conducted with $Ti(OBu)_4$ (0.05 mol% resp. diacid, added in two portions after 0 h and 5 h esterification), BuSnOOH (0.05 mol% resp. diacid) was added after 9 h esterification. It was observed in previous experiments that the dark brown colouration in reactions with guaiacol only appears with BuSnOOH catalyst after ~5 h esterification.

The mechanical properties in this series of polyesters (Fig. 5b,c) depend on the diacid monomer, caused by differences in rigidity, and thus $T_g$ values, of the respective materials. PIsSu and poly(isosorbide diglycolate) (PIsDga) exhibited the most interesting properties. Both materials are very rigid (as indicated by the high Young's Modulus of up to 3870 MPa) and have a higher ultimate tensile strength than established high $T_g$ polymers such as Eastman's Tritan, a fossil-based high-performance polyester, or the biobased poly(lactic acid) (PLA)[20,21]. A comparison of PIsSu with a recently reported super engineering thermoplastic, based on isosorbide and 4,4'-difluorodiphenyl sulfone (SUPERBIO, $T_g = 212$ °C)[22], reveals comparable (with regard to Young's modulus) or superior properties (ultimate tensile strength and elongation at break) for PIsSu.

Similarly interesting trends were found for the barrier properties of some of the synthesized materials (Fig. 5d). The oxygen barrier of PIsSu, PIsDga, and poly(isosorbide glutarate) (PIsGlu) was found to be superior to PET. The lowest oxygen permeability (OP) was found for PIsDga (0.24 mm cm³ m⁻² day⁻¹ bar⁻¹), which is comparable to high barrier materials, such as the recently reported poly(lactic-co-glycolic acid) (PLGA) copolymers (>80% glycolic acid content, OP < 1 mm cm³ m⁻² day⁻¹ bar⁻¹)[23]. The water barrier of all investigated materials was somewhat inferior to PET and PLGA copolymers. Superior material properties of new biobased materials are important for their industrial adaptation, as they can incentivise the subsequent replacement of fossil-based materials[2], rather than drop-in materials that can only compete on price.

## Scale-up in 2 L autoclave

To prove the industrial viability of this method, the synthesis of PIsSu was scaled up from a 100 mL glass reactor to a 2 L stainless steel autoclave (see Supplementary Figs. 50 and 51 for photos of the respective reactors) under optimized reaction conditions. The reaction was conducted on a 3.1 mol scale, which corresponds to 366 g of succinic acid, 453 g of isosorbide, and 503 g p-cresol. The ring opening hydration of isosorbide to 1,4-sorbitan, previously described by Yoon et al. (Fig. 1b)[24], was observed in reactions conducted in the 2 L autoclave. This led to a small excess of alcohol end groups in oligomers after esterification, which, as previously mentioned, reduces the molecular weight of the final product. To account for this, a 1.5 mol% excess of succinic acid was added during esterification (corresponding to mol% of 1,4-sorbitan calculated by ¹H NMR spectrum of esterification sample, see also Supplementary Figs. 54, 55). High molecular weight PIsSu was obtained after 1 h full vacuum polycondensation in the autoclave ($M_n = 35.3$ kg/mol, PDI = 2.2, $T_g = 80$ compared to $M_n = 38.9$ kg/mol, PDI = 2.0, $T_g = 81$ °C obtained in glass). Around 500 g of PIsSu were obtained within 1 h of extrusion at 3 bar and 220 °C, which corresponds to a yield of 70% (Supplementary Fig. 57, $m_{theoretical}$(PIsSu) = 707 g). More polymer could have been obtained, but extrusion becomes increasingly slow during its later stages. The extruded and chipped product was comparable to PIsSu synthesized in a glass reactor regarding mechanical and barrier properties (Supplementary Tables 7 and 8).

## Biodegradability and hydrolytic stability of PIsSu

When designing new biobased materials it is important to consider their entire life cycle, including options for reuse, (chemical) recycling and ultimately the material's fate in the environment[2]. The biodegradability and hydrolytic stability of PIsSu under ambient conditions was investigated (see Supplementary Figs. 58 and 59). The former was studied by following $CO_2$ evolution in soil burial experiments, in which no significant degradation was observed after more than a year. The hydrolytic stability was followed by ¹H NMR at neutral pH and no soluble compounds were identified after 195 days at 25 °C. This contradicts previous reports where PIsSu was found to be degradable under soil burial and enzymatic hydrolysis conditions[12,25]. This difference could be due to the significantly lower molecular weights and higher temperature used in those studies ($M_n$ ~7.5 kg/mol; measurements conducted at 37 °C) and it is very well possible that given more time the higher molecular weight materials from the present study will also show biodegradation[26]. The degradability characteristics of high molecular weight polyesters based on isosorbide do, however, require further investigation. Similar to other polyesters, the ester functionality in the presented polymers enables chemical recycling[4].

## Discussion

In conclusion, we presented a synthetic method for the synthesis of high molecular weight polyesters based on the biobased, yet unreactive secondary diols isosorbide and isomannide. The in situ formation of reactive aryl esters during esterification leads to a significant enhancement of the end group reactivity during polycondensation and enables the synthesis of high molecular weight materials with 100 mol% biobased, rigid secondary diol content. To date, these materials have been impossible to synthesize with molecular weights high enough for applications that require ductile properties. The described method is applicable to both known and novel polyester compositions. The described isosorbide-based materials show promising barrier and mechanical properties that can outperform common fossil-based materials. The operational simplicity and use of standard polyester synthesis equipment and monomers could spark further research into applications of the polymers that are now accessible. Other, previously inaccessible, polyester compositions

**Table 5 | Polyesters synthesized with isosorbide, isomannide and aliphatic diacids by *p*-cresol assisted polyesterification**

| Polymer | $M_n$ [kg/mol] | PDI | $T_g$ [°C] | Polymer | $M_n$ [kg/mol] | PDI | $T_g$ [°C] |
|---|---|---|---|---|---|---|---|
| Poly(isosorbide glutarate) (PIsGlu) | 41.0 (16.0[a, b, c])[25] | 2.1 (1.7) | 52 (28) | Poly(isomannide succinate) (PImSu) | 28.3[h] (29.0[a, b, c])[28] | 2.8 (2.1) | 82[i] (82) |
| Poly(isosorbide adipate) (PIsAd) | 29.4 (10.1[d, e])[13] | 2.1 (2.7) | 35 (20) | Poly(isomannide glutarate) (PImGlu) | 40.1 (11.0[a, b, c])[25] | 2.7 (1.6) | 51 (37) |
| Poly(isosorbide-1,4-cyclohexanedicarboxylate) (PIsCyc) | 40.1 (18.3[f])[24] | 2.2 (4.6) | 133[g] (128) | Poly(isomannide adipate) (PImAd) | 30.2 (20.0[a, b, c])[25] | 2.2 (1.5) | 35 (28) |
| Poly(isosorbide diglycolate) (PIsDga) | 22.3 (/) | 1.9 | 83 | Poly(isomannide-1,4-cyclohexanedicarboxylate) (PImCyc) | 32.6 (/) | 2.4 | 134[j] |
| Poly(isosorbide thiodiglycolate) (PIsThd) | 16.9 (/) | 1.8 | 57 | Poly(isomannide diglycolate) (PImDga) | 20.4 (/) | 2.0 | 80 |

Previously reported literature values for molecular weights and $T_g$ are given in brackets for known polyester compositions. If no values are indicated, the respective polyesters have not been reported. For detailed synthesis conditions, see Supplementary Table 1.
[a]synthesized from the respective diacid chloride.
[b]determined after precipitation in $CH_3Cl/MeOH$.
[c]Polystyrene was used as a SEC calibration standard.
[d]Poly(methyl methacrylate) was used as a SEC calibration standard.
[e]Synthesized from dimethyl adipate. Synthesis from adipoyl chloride: 26.0 kg/mol, PDI =1.5, $T_g$ = 40 °C[29].
[f]Synthesized with 30 mol% (respective isosorbide) of acetic anhydride, hence the high PDI.
[g]*cis/trans* ratio of 1,4-cyclohexanedicarboxylic acid: 41.8/58.2.
[h]Polymer was melt quenched in liquid nitrogen to enable dissolution in DCM.
[i]This polymer is the only semi-crystalline material synthesized in this work. Melting points coincided with previous literature reports[28]. $T_c$ = 131 °C, $T_{m1}$ = 172 °C, $T_{m2}$ = 185 °C.
[j]*cis/trans* ratio of 1,4-cyclohexanedicarboxylic acid: 39.4/60.6.

based on monomers with a low reactivity as well as the application of similar methods in other classes of step growth polymers (polyamides, polycarbonates) could be explored further.

## Methods
### Materials
Regular purity isosorbide was purchased from Carbosynth (>98%) and purified further by recrystallization from acetone and subsequent distillation over $NaBH_4$. High-purity isosorbide was purchased from Roquette Fréres (Polysorb grade, ≥99.9%). Succinic acid was purchased either from Acros Organics (>99%, regular purity) or from Thermo Scientific (≥99%, high purity). Unless otherwise noted in Table 4, the normal purity compounds were used for the synthesis of polyesters. Isomannide was purchased from Carbosynth (>98%) and purified by recrystallization from acetone. *p*-Methyl phenol (*p*-cresol, >99%), *o*-methoxy phenol (>99%), adipic acid (>99%), diglycolic acid (>98%), thiodiglycolic acid (>98%) and germanium dioxide (99.9%)

were purchased from Acros Organics. Butyltin hydroxide oxide hydrate (97%), titanium(IV) butoxide (>97%), zinc(II) acetate (>99.9%), glutaric acid (>99%), 1,4-cyclohexanedicarboxylic acid (>99%), 1,4-dimethoxybenzene (>99%), diphenyl ether (>99%) and cellulose (powder, 20 μm average particle size) were purchased from Sigma Aldrich. *p*-Methoxy phenol (>99%) was purchased from Fisher Scientific. Petraerythritoltetrakis(3-(3,5-di-t-butyl-4-hydroxyphenyl)propionate) (Irganox 1010) (98%) and Tris(2,4-di-tert-butylphenyl)phosphite (Irgafos 168) (98%) were purchased from ABCR. Zirconium(IV) butoxide (80% w/w in 1-butanol) was purchased from Alfa Aesar. Acetone and toluene were purchased from VWR. DMSO-d6, DCM-d2, and $D_2O$ were purchased from Sigma Aldrich. Soil (LUFA 2.2) was obtained from LUFA Speyer.

### Characterization and processing techniques
[1]H NMR spectra were recorded on a Bruker AMX 400, [13]C NMR, and 2D spectra were recorded on a Bruker DRX 500. Spectra were referenced

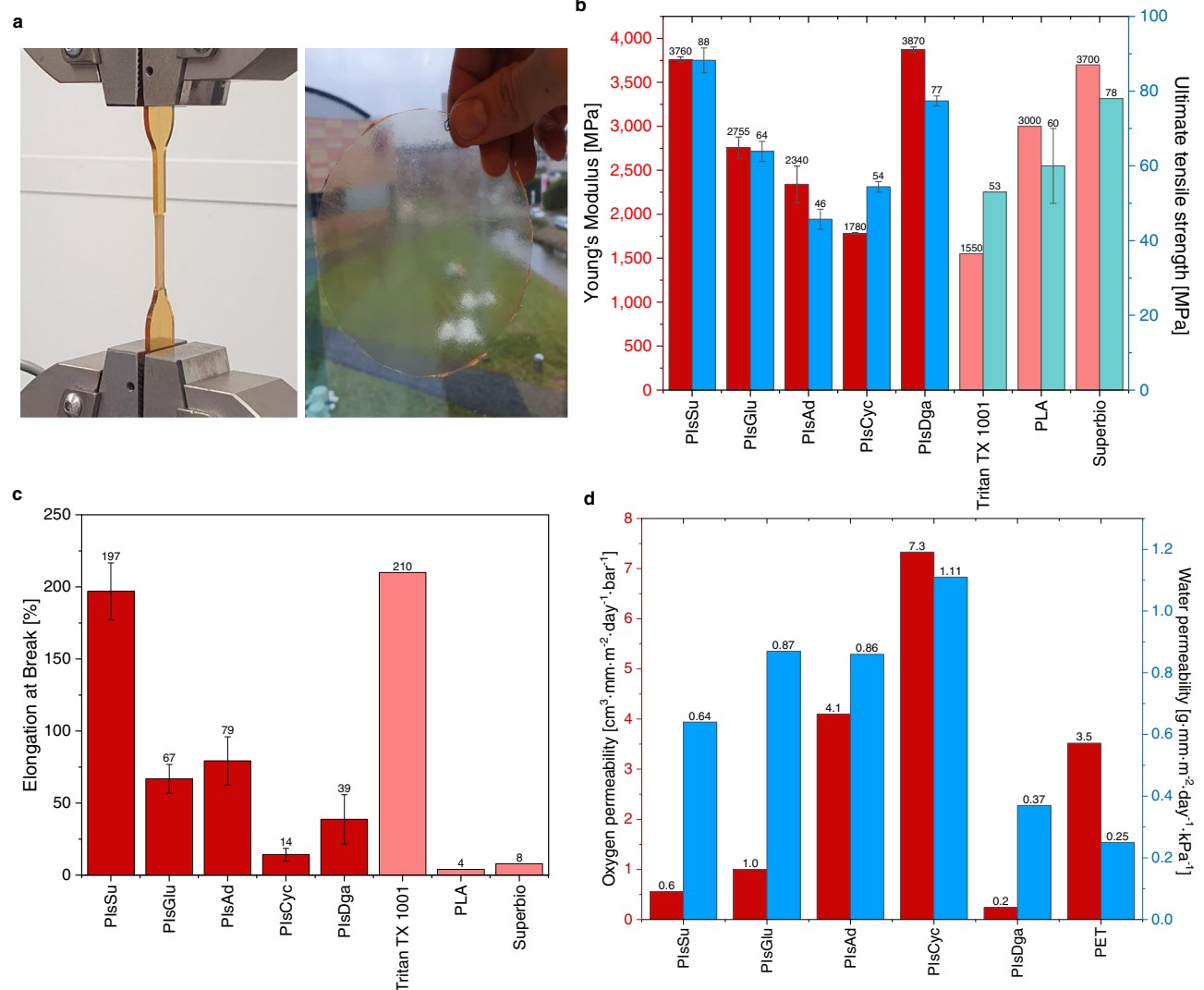

**Fig. 5 | Barrier and mechanical properties of isosorbide-based polyesters. a** Pictures of a PIsSu tensile bar during tensile testing and a PIsSu polymer film. **b** Young's Modulus and ultimate tensile strength of isosorbide-based polyesters. Shown values are averaged from three measurements. PIsSu values were determined with samples synthesized in the 2 L autoclave. Reference values for Tritan TX 1001, PLA and Superbio were taken from the literature[20–22]. **c** Elongation at break of isosorbide-based polyesters. Shown values are averaged from three measurements. **d** Oxygen and water permeability of isosorbide-based polyesters. The oxygen permeability was determined at 30 °C and 50% humidity. The water permeability was determined at 38 °C and 90% humidity (PIsAd was measured at 30 °C due to the material's low $T_g$). The values for PET were determined with commercially available PET (RamaPET N180, 10% crystallinity, not biaxially oriented).

to the residual solvent signal. For $^1$H, $^{13}$C, and 2D NMR spectra of all synthesized polyesters, see Supplementary Figs. 6–42.

Molecular weights were measured using Size Exclusion Chromatography (SEC) on an Agilent HPLC system (1260 Infinity II) with two PLgel 5 µm MIXED-C (300 × 7.5 mm) columns and a 1260 Infinity II Refractive Index detector. Dichloromethane was used as mobile phase with a flow of 1 mL/min at T = 35 °C. A sample concentration of 5 mg/ml was used, the injection volume per sample was 50 µl. Polystyrene standards ($M_n$ = 550 g/mol to 6,025,000 g/mol, PS-H Easy Vial from Sigma Aldrich) were used for calibration. Calculation of the molecular weights was carried out with Agilent GPC/SEC software for OpenLAB CDS.

Differential scanning calorimetry (DSC) measurements of polymers were carried out on a DSC 3+ STARe system from Mettler Toledo. The polymer sample (3–5 mg) was sealed in an aluminium pan (40 µm) and subjected to two subsequent heating and cooling cycles (heating/cooling rate 10 K min⁻¹) from 0 to 300 °C under a constant nitrogen flow (50 mL min⁻¹). Data reported was taken from the second heating cycle. For

DSC traces of all synthesized polyesters see Supplementary Figs. 43–45.

Thermogravimetric analysis (TGA) measurements were carried out on a TGA/DSC 3+ STARe system from Mettler Toledo. The polymer sample (20–25 mg) was sealed in an aluminium pan (100 µm). The sample was then heated from 25 °C to 500 °C at 10 K min⁻¹ under a constant nitrogen flow (50 mL min⁻¹). See Supplementary Fig. 46 for an overlay of all TGA traces and Supplementary Table 3 for $T_{5\%d}$ values.

Tensile bars were produced with a Thermo Scientific HAAKE Minijet II apparatus according to ISO-527-2, sample type A5. The polymer samples were dried overnight in a vacuum oven before injection moulding. Around 2 g of polymer material were used per tensile bar. An injection pressure of 1200 bar was used and held for 8 s after injection. Mould and cylinder temperatures were varied depending on the polymer. For the exact conditions for each polymer, see Supplementary Table 5.

The tensile properties were determined on a Instron 5565 machine with a load cell (10 kN) and an Instron strain gauge extensometer (2630-106-25 mm). Sample size was set to width (4 mm),

thickness (1.95 mm), parallel length (20 mm), and test speed (5 mm/s). The extensometer was removed at an extension of 12 mm and the measurement was continued after. All samples were tested until rupture at this same speed. See Supplementary Fig. 49 for stress-strain graphs.

Polymer films were prepared by compression moulding using a thermal press (Carver Auto Four/3015-NE,H). Around 2 g of dried polymer was sandwiched between two poly(tetrafluoroethylene) (PTFE) films (0.14 mm thickness) and two aluminum plates (3 mm thickness). The plates with the polymer were placed upon the pre-heated bottom plate of the thermal press. See Supplementary Table 4 for the temperatures used for each polymer. After -1 minute, the polymer was softened and a pressure of 0.5 ton was applied. The pressure was held for 15 s and subsequently increased to 1, 2.5, 5, 10, and 20 tons in 15 s intervals. After 20 s at 20 tons, the pressure was released and the average thickness of the polymer film was measured with an electronic micrometer.

The oxygen and water permeability was determined using a Totalperm (Permtech s.r.l) instrument. Calibration of the system was carried out with a standard PET film provided by Permtech (Italy), according to the ASTM F1927-14 standard. The system reports the oxygen transmission rate (OTR) at the established conditions; these values were normalized for the film thickness (x) to determine the oxygen permeability. The water permeability was measured using the same Totalperm (Permtech s.r.l) instrument calibrated for water vapor according to the ASTM E96/E96M-15 standard. The measurements were complete when the collected data had reached a tolerance level of 0.5%. Water vapour transmission rate values were normalized by the film thickness and divided by the water saturation pressure at the indicated temperature and relative humidity (in kPa) to obtain water permeability values. See Supplementary Figs. 47 and 48 for raw data graphs.

A Respicond respirometer was used for determining the biodegradability of PIsSu, details of the method were described in a recent publication[27]. Briefly, biodegradation tests were performed in the dark, in closed 250 mL vessels which were maintained at 25 °C. $CO_2$ evolved from the test medium was trapped by a potassium hydroxide solution inside the vessel. Then the amount of trapped $CO_2$ was calculated based on the decrease in the conductivity of the KOH solution. Conductance in the KOH solution was measured hourly and the solution was refreshed regularly, before the absorption of $CO_2$ reached its limit. The incubation experiments lasted ~410 days, including two abiotic controls, five replicates of blanks (soil without test material), triplicates of cellulose (positive reference), and five replicates of test material (PIsSu). Polymers were ground and sieved through a 600 μm mesh filter (except cellulose powder added as obtained). Typically, ~120–170 mg test material (equivalent to 75 mg of carbon) was added on top of around 19 g wet soil (equivalent to 15 g dry soil) in each vessel. It means the carbon loading was kept 5 mg C g⁻¹ dry soil in all experiments with test materials.

Hydrolytic stability of PIsSu was measured in an NMR tube. Approximately 10 mg polymer powder (<600 μm) was added to 1 mL $D_2O$, containing 3.6 mM dimethyl sulfoxide (DMSO) as a standard. These tubes were subsequently kept at a controlled temperature of 25 °C. The hydrolysis experiment was performed over 195 days. Samples were typically measured every one or two weeks. An Avance III 400 MHz NMR spectrometer ($^1$H NMR) was used to measure soluble hydrolysis products. All monomers resulting from polymer hydrolysis are soluble in $D_2O$ and can therefore be quantified.

### Polymerization experiments in 100 mL glass reactor
Isosorbide (8.768 g, 60.0 mmol, 1.0 eq.), succinic acid (7.085 g, 60.0 mmol, 1.0 eq.), and p-cresol (9.732 g, 90.0 mmol, 1.5 eq.) were weighed into a three-necked 100 mL round bottom flask. Butyltin hydroxide oxide hydrate (0.013 g, 0.06 mmol, 0.001 eq.) was added to

the reactor as a suspension in 0.5 mL toluene. The flask was equipped with a nitrogen gas inlet, a top stirrer, and a distillation head with a thermometer and a receiving flask attached to it. The reactor was heated to 240 °C under a constant nitrogen flow. Stirring was initiated at a rate of 100 rpm when the oil temperature reached 150 °C. After 5 h at 240 °C a melt sample of the reaction mixture was taken under a positive nitrogen flow to check the conversion. The oil was then cooled to 220 °C and a rotary vane oil pump was connected to the receiving flask to initiate pre-polycondensation. A vacuum of 400 mbar was slowly applied and held for 15 min. The pressure in the reactor was halved (200, 100, 50, 25, 12.5, and 6.5 mbar) every 15 min until a pressure of 0.4–0.8 mbar was reached. The stirring speed was then reduced to 30 rpm due to the viscous polymer melt. Polycondensation was continued for 1 h, when, in successful reactions with a sufficiently high molecular weight polymer, stirring of the reaction melt became impossible without breaking the vacuum due to the highly viscous polymer melt. The reactor was flushed with nitrogen and the polymer product was scraped from the reactor with a spatula under a positive nitrogen flow. This synthesis procedure was adjusted depending on the diol and diacid monomers used: See Supplementary Table 1 for the exact conditions for each polyester composition.

### Polymerization experiments in 2 L stainless steel autoclave
The synthesis was conducted with high-purity monomers (see Materials). Isosorbide (453.0 g, 3.1 mol, 1.0 eq.), succinic acid (366.1 g, 3.1 mol, 1.0 eq.), p-cresol (502.8 g, 4.65 mol, 1.5 eq.), butyltin hydroxide oxide hydrate (0.647 g, 3.1 mmol, 0.001 eq.) and tris(2,4-di-tert-butylphenyl) phosphite (0.351 g, 0.5 mmol, 0.0002 eq.) were weighed into a 2 L stainless steel autoclave. The reactor was closed and heated to 220 °C under a constant nitrogen flow. Stirring was initiated at a speed of 100 rpm when the oil temperature reached 150 °C. After 1 h at 220 °C, the oil temperature was increased to 240 °C. This temperature was held for 5 h until no more water was collected in the receiving flask of the reactor (see Supplementary Fig. 52). A melt sample of the reaction mixture was taken under a positive nitrogen flow to quantify the alcohol to ester end group ratio and the amount of 1,4-sorbitan (Supplementary Fig. 55). Based on that observation, succinic acid (5.49 g, 46.5 mmol, 0.015 eq.) was added to the reaction mixture and stirred for another 1.5 h at 240 °C. After, the oil temperature was decreased to 220 °C and pre-polycondensation was initiated by slowly applying a vacuum of 400 mbar. The pressure in the reactor was halved (200, 100, 50, 25, 12.5, and 6.5 mbar) every 15 min until a pressure of 0.1–0.5 mbar was reached. The reaction was deemed complete when a torque of around 1200 Ncm⁻¹ was reached, which typically took between 1 and 1.5 h at 0.05–0.3 mbar. For a detailed plot of reactor parameters (p, T, torque, rpm) see Supplementary Fig. 56. The reactor was then flushed with nitrogen and a pressure of 3 bar was applied to the reactor. The polymer product was then extruded through the bottom nozzle of the reactor into a water bath and chipped.

## Data availability
The authors declare that the data supporting the findings of this study are available within the article and its Supplementary Information file. All other relevant source data is available from the corresponding author upon reasonable request.

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

## Acknowledgements

This research was funded by the Dutch Research Council (NWO), [grant number 731.017.203]. The authors would like to thank LEGO SYSTEM A/S and Avantium Chemicals B.V. for co-funding this project. The authors would like to thank Dr. Rene Dam for the helpful discussions around scale-up experiments. We would also like to thank Dr. Andreas Ehlers for his assistance with NMR measurements.

## Author contributions

G.J.M.G., B.W., and D.H.W. conceived the concept. D.H.W., B.W., and R.J.P. devised the experimental program. D.H.W. performed experiments on polyester synthesis and its property characterization. Y.W. conducted experiments on hydrolysis and biodegradability. D.H.W., G.J.M.G., and R.J.P. wrote the manuscript. B.B.P. and K.M. assisted with the preparation of samples for mechanical testing and the analysis of the mechanical data.

## Competing interests

A patent application (EP21217721), with G.J.M.G., B.W., D.H.W., R.J.P., B.B.P. as inventors, has been filed by Avantium and the University of Amsterdam on findings reported in this work. The authors of this work declare no competing interests.
