## [Peer Review File · Nature Communications]

REVIEWER COMMENTS

Reviewer #1 (Remarks to the Author):

This manuscript reports advances in isosorbide-based polyesters. Although isosorbide as an available biobased diol has been much studied for polyesters, applications have been hindered by limited molecular weights and mechanical properties. By clever adaption of a procedure described for PET (as the authors point out), using p-cresol as a reactive solvent component reasonable molecular weights that result in ductile mechanical properties are achieved. The p-cresol is shown to function by generating reactive endgroups which enhance molecular weight build-up in the final stages of polymerizations, the reagent itself being released and recycled. Overall, this might be considered rather a technical improvement for a particular system. On the other hand, this work addresses a very relevant challenge. If the practical application of isosorbide for polyesters can be enabled, this is of strong interest for a broad scientific community.

To underline the advance made with this method the authors comprehensively compare molecular weight data to previously reported literature values. This important discussion is largely meaningless in its present state as the reference points are not given and this will easily make a difference of a factor two to three regarding the true molecular weights. For all literature molecular weights listed, details should be specified (for GPC molecular weights, the exact calibration methods and standards).

More troubling, the authors fail to specify the calibration method for any of the extensive GPC data reported. Not even the GPC instrumentation is listed, unlike all other methods.

In the discussion of the scope of the method, rather than different diacids the scope regarding other secondary diols would be of interest as it is the isosorbide diol monomer that hinders the polymerization.

The very brief section on 3D printing appears preliminary and should be eliminated unless substantiated with significant data, e.g. comparing the quality of printed specimens to other materials in detail concerning mechanical properties or accuracy.

Chemical recycling in a header raises expectations on data and results to this end, however, the authors only point out that as a polyester the material should be recyclable without any experiments to this end.

Further comments:

Referring to Global Warming and the Plastics Crisis in the opening sentence appears superficial and generic rather than convincing given this is not followed up in any way in the manuscript.

In the Extended Data Figure 2, labelling (1) and (2) is used for the kresol as well as ester endgroups. Further, using a consistent labelling scheme would be helpful (e.g. Extended Data Figure 2 vs. 3).

In the Extended Data Figure 2, the data refers to 'total unreacted isosorbide (respective total mol% of isosorbide'. Is this truly unreacted isosorbide (how was this determined), or unreacted -OH groups?

Extended Data Figure 3. A water layer is referred to, which is not discussed otherwise in the manuscript. The formation of water and its removal, also the removal from recycled kresol, should be discussed.

The discussion of the choice of reagents overall is not substantiated by data. For example, the statement 'Firstly, the aryl alcohol should have a sufficiently high ambient pressure boiling point[12] to enable esterification at 240 °C.' This sounds reasonable, but no relevant data is shown to underline this suggestion. Also the discussion of substitution is purely phenomenological and based on two examples.

Rather than subscripts such as 'Reaction conditions (t, T, p) were the same for all experiments unless noted otherwise', listing the conditions would much improve readability.

In the discussion of scale-up, rather than noting the reactor vessel size the amount of monomer used and polymer produced appears of interest. Currently this data is to be found only in the Supporting Information.

Giving DSC data to one digit certainly exceeds the accuracy and reproducibility of the method.

Reviewer #2 (Remarks to the Author):

This manuscript by D. H. Weinland et al. reported a new synthetic method for the synthesis of high molecular weight polyesters based on the biobased, yet unreactive secondary diols isosorbide and isomannide by the in situ formation of reactive aryl esters during esterification. It seems that the authors tried their best efforts to prove their concept. And scientific flow is rationale. However, unfortunately, the reviewer couldn't find enough novelty for the publication in Nature Communications because of followings:

1. It is good to use p-cresol for the enhanced reactivity for efficient polycondensation of bio-based isosorbide/isomannide monomers. But this concept is already published previously in *Macromolecules* ([dx.doi.org/10.1021/ma4001435](https://doi.org/10.1021/ma4001435)). The authors can point out the difference including PDI values and reusability of byproducts, but these facts can attract interests from polymer-specialized society.

2. Poly(isohexide succinate) is commonly known already ([dx.doi.org/10.1021/acs.macromol.8b02594](https://doi.org/10.1021/acs.macromol.8b02594)), and data presented by the authors is also ordinary. It is difficult to expect more interesting characteristics from this polymer. Moreover, unfortunately, the authors didn't seem to find remarkable properties that may attract attention.

- The authors didn't provide mechanical data of 3D printed objects.

- In the section of Chemical recycling and end of life, data is insufficient to discuss the meaning for sustainability.

Minor points:

-No proper abstract was provided.

-In Fig. 1, illustrated isosorbide repeating unit in the polymer chain must lose the chirality. It should be revised.

Reviewer #3 (Remarks to the Author):

The manuscript by Weinland et al. is an interesting report of improving molecular weight and thus properties in isosorbide monomers via step growth polymerization. Overall, the data is compelling and important to the field of sustainable polymers and this reviewer enjoyed reading the article and supportive of publication with revisions. There are a few things below that should be addressed prior to publication:

- 1) What is the mole ratio of p-cresol?
- 2) Do the kinetics follow Carother's equation?
- 3) What is the typical T_gs of the lower molecular weight isosorbide esters (it would be good to report these for comparison)?
- 4) What is the price point of the final product and how does it compare to current commercial polymers (particularly isosorbide-containing polymers) used in industrial products?
- 5) Why are the PDIs so high (>2.0), which is beyond Carother's theory for step-growth at 100% conversion? Are there side reactions occurring for these polymers?
- 6) Generally, the figures need work and quality improved for publication.
- 7) Why is the material yellow-brown in color (is it partly degrading during the printing)?

REVIEWER COMMENTS

Reviewer #1 (Remarks to the Author):

This manuscript reports advances in isosorbide-based polyesters. Although isosorbide as an available biobased diol has been much studied for polyesters, applications have been hindered by limited molecular weights and mechanical properties. By clever adaption of a procedure described for PET (as the authors point out), using p-cresol as a reactive solvent component reasonable molecular weights that result in ductile mechanical properties are achieved. The p-cresol is shown to function by generating reactive end groups which enhance molecular weight build-up in the final stages of polymerizations, the reagent itself being released and recycled. Overall, this might be considered rather a technical improvement for a particular system. On the other hand, this work addresses a very relevant challenge. If the practical application of isosorbide for polyesters can be enabled, this is of strong interest for a broad scientific community.

To underline the advance made with this method the authors comprehensively compare molecular weight data to previously reported literature values. This important discussion is largely meaningless in its present state as the reference points are not given and this will easily make a difference of a factor two to three regarding the true molecular weights. For all literature molecular weights listed, details should be specified (for GPC molecular weights, the exact calibration methods and standards).

An important oversight on our side, thank you for pointing this out. Indeed we forgot to include the SEC method in our manuscript. We included the method and the calibration method and standards in the revised draft.

We also included information on the respective calibration standards used to obtain the cited literature values. We could however not find detailed information on the exact calibration methods used in any of the references (except for Ref. 20 (Yoon *et al.*) who mentions that a universal calibration was performed). In our case, calibration is performed with pre-purchased, standardized polystyrene standards in a twelve point calibration. Apart from GPC molecular weights, T_g values can also be an indication if a high molecular weight material is obtained. At low molecular weights, T_g values depend on the molecular weight, as can be seen in Fig. 1 (formerly Extended Data Fig. 1a), where a large variation of T_g values for PIsSu indicates that a high enough molecular weight to reach a plateau value for the T_g has thus far not been reached (compare also the T_g value of 82 °C we measured for PIsSu to the highest previously reported T_g value of 73 °C).

More troubling, the authors fail to specify the calibration method for any of the extensive GPC data reported. Not even the GPC instrumentation is listed, unlike all other methods.

See above.

In the discussion of the scope of the method, rather than different diacids the scope regarding other secondary diols would be of interest as it is the isosorbide diol monomer that hinders the polymerization.

An interesting point when considering the traditional approach to a new synthesis/catalysis method report in (organic) chemistry. We do believe however that in this case the range of secondary diols that are available on a commercial scale is quite limited. Some further elaboration on this:

1. To our knowledge only 2,2,4,4-tetramethylcyclobutanediol and isosorbide are secondary diols commercially available at scale. The former is fossil-based and preliminary results from our group indicate a relatively low stability of that monomer under acidic conditions (free carboxylic acids and even weak acids like aryl alcohols). There are other secondary diols available from chemical suppliers, such as 1,4-cyclohexanediol or hydrogenated bisphenol A, although these are currently not used in commercial polyester application to our knowledge. We wanted to focus our attention in the present manuscript on the new synthesis method using aryl alcohols in combination with sustainably sourced monomers.
2. The amount of different polymers reported in our paper is already significantly larger than commonly reported in polymer chemistry papers (ignoring the fact that varying diol amounts in copolyesters is not really reporting 'new' polyesters). Considering the limited availability of other secondary diols for polyester synthesis, the authors do not see an immediate benefit to the content of the paper. We do believe however that future work could be conducted on other polyesters based on unreactive secondary diols that are currently not commercially available (see 1.) and aliphatic diacids.

The very brief section on 3D printing appears preliminary and should be eliminated unless substantiated with significant data, e.g. comparing the quality of printed specimens to other materials in detail concerning mechanical properties or accuracy.

We understand the feedback. We did not do any comparisons regarding properties of 3D printed specimen. A detailed comparison would take additional experiments on 3D printing and mechanical properties, which is not planned for the current manuscript. We also believe that it is out of the scope of the manuscript, as the new synthesis method using aryl alcohols, together with it enabling access to previously unattainable materials, is its main focus. In our opinion, more details on 3D printing would be lost in the already densely packed manuscript. The reported 3D printing results were included to show a potential (high value) application of PIsSu. Further data and characterization of 3D printed specimen will be reported elsewhere. If the reviewer disagrees with this, 3D printing results can be removed upon request.

Chemical recycling in a header raises expectations on data and results to this end, however, the authors only point out that as a polyester the material should be recyclable without any experiments to this end.

Indeed no chemical recycling experiments have been reported. We removed chemical recycling from the header and rephrased the respective passage in the revised manuscript pointing to the general chemical recyclability of polyesters.

Further comments:

Referring to Global Warming and the Plastics Crisis in the opening sentence appears superficial and generic rather than convincing given this is not followed up in any way in the manuscript.

We adjusted this in the introduction of the revised manuscript.

In the Extended Data Figure 2, labelling (1) and (2) is used for the kresol as well as ester endgroups. Further, using a consistent labelling scheme would be helpful (e.g. Extended Data Figure 2 vs. 3).

We adjusted the labelling to be consistent throughout the manuscript.

In the Extended Data Figure 2, the data refers to 'total unreacted isosorbide (respective total mol% of isosorbide'. Is this truly unreacted isosorbide (how was this determined), or unreacted -OH groups?

We refer to unreacted isosorbide as determined by ^1H NMR. We referred to this in the caption of Fig. 3 (formerly Ext Dat Fig. 2). We made it more clear in the revised draft and included the formula used to calculate total free isosorbide in the SI.

Extended Data Figure 3. A water layer is referred to, which is not discussed otherwise in the manuscript. The formation of water and its removal, also the removal from recycled kresol, should be discussed.

We included a discussion of this in the revised manuscript. Water is formed by the reaction of diol/aryl alcohol with diacid. In the recycling experiments, no separation was conducted, as addition of water to the reactants did not seem to hamper the reaction. The water can easily be separated by phase separation (apart from small amounts dissolved in cresol) in larger scale experiments.

The discussion of the choice of reagents overall is not substantiated by data. For example, the statement 'Firstly, the aryl alcohol should have a sufficiently high ambient pressure boiling point[12] to enable esterification at 240 °C.' This sounds reasonable, but no relevant data is shown to underline this suggestion. Also the discussion of substitution is purely phenomenological and based on two examples.

A valid point of concern. To improve the clarity on this, we added a more in-depth discussion on the criteria of aryl alcohol choice in the manuscript. This also extends now to the substitution pattern of the aryl alcohols in question. In the present work we wanted to focus on industrially feasible aryl alcohols and we believe that the investigated derivatives are representative of a larger group of phenols. Within our group, a more detailed investigation on the influence of aryl alcohol substitution has been done, but these experiments were performed for a different polyester composition and will be published elsewhere.

Apart from that, we included a more in-depth discussion of the optimization of reaction conditions due to more available space (the initial Nature journal the manuscript was submitted to had more stringent word count limitations).

Rather than subscripts such as 'Reaction conditions (t, T, p) were the same for all experiments unless noted otherwise', listing the conditions would much improve readability.

A good point, we included the exact conditions in the relevant figures.

In the discussion of scale-up, rather than noting the reactor vessel size the amount of monomer used and polymer produced appears of interest. Currently this data is to be found only in the Supporting Information.

A good point, we included this in the main text.

Giving DSC data to one digit certainly exceeds the accuracy and reproducibility of the method.

We adjusted the reported T_g values.

Reviewer #2 (Remarks to the Author):

This manuscript by D. H. Weinland et al. reported a new synthetic method for the synthesis of high molecular weight polyesters based on the biobased, yet unreactive secondary diols isosorbide and isomannide by the in situ formation of reactive aryl esters during esterification. It seems that the authors tried their best efforts to prove their concept. And scientific flow is rationale. However, unfortunately, the reviewer couldn't find enough novelty for the publication in Nature Communications because of followings:

1. It is good to use p-cresol for the enhanced reactivity for efficient polycondensation of bio-based isosorbide/isomannide monomers. But this concept is already published previously in *Macromolecules* ([dx.doi.org/10.1021/ma4001435](https://doi.org/10.1021/ma4001435)). The authors can point out the difference including PDI values and reusability of byproducts, but these facts can attract interests from polymer-specialized society.

We believe that the approach reported here using p-cresol differs in several ways from the cited publication by Yoon *et al.*:

1. PDI of polymers significantly changes their mechanical- and thermal properties as well as processing conditions. PDI values reported by Yoon *et al.* reach up to 5.6 (if a M_n of 17.1 kg/mol is targeted, $T_g = 124$ °C), which is very high compared to typical PDI values of polyesters (PDI < 3). This is also reflected by the large variation in T_g values of poly(isosorbide-1,4-cyclohexanedicarboxylate) reported by Yoon *et al.* These high PDI values are caused by the presence of significant amounts of 1,4-sorbitan, which is formed by the acetic acid catalysed ring-opening hydration of isosorbide. This 1,4-sorbitan is incorporated in the polymer chain and also influences the properties of the polymer product. Formation of 1,4-sorbitan and subsequent branching of the polymer does not happen in the present aryl alcohol assisted synthesis method. For example we obtained a PDI of 2.2 for the same polymer, poly(isosorbide-1,4-cyclohexanedicarboxylate), while reaching M_n values of 40.1 kg/mol ($T_g = 133$ °C). One can argue if the same polymer material is obtained in the end due to the reasons stated above.
2. We have shown in our manuscript that the aryl alcohol assisted synthesis is applicable to a range of polyester compositions. The procedure reported by Yoon *et al.* is only applicable to the described poly(isosorbide-1,4-cyclohexanedicarboxylate). Attempts to utilize acetic anhydride in the synthesis of other isosorbide-based polyesters did not result in any reactivity enhancement according to a subsequent study published by the authors (<https://pubs.acs.org/doi/10.1021/ma4015092>).
3. As mentioned already, our approach results in polymer products with low PDI values while utilizing a reusable reagent. Acetic anhydride is a substoichiometric reagent that needs to be chemically regenerated. This is important considering potential scale-up of the described polymer products, as acetic acid (the by-product of the procedure reported by Yoon *et al.*) can only be converted back to acetic anhydride under severe reaction conditions ($T \sim 700$ - 800 °C), which makes the process expensive, requiring special equipment and safety considerations. p-Cresol on the other hand can be reused directly, as we have shown in our manuscript, while obtaining higher M_n values with lower PDI values.

For these reasons, we believe that the aryl alcohol assisted synthesis of polyesters based on unreactive secondary diols like isosorbide presents a novel approach with clear advantages over the previously described synthesis approach using acetic anhydride.

2. Poly(isohexide succinate) is commonly known already ([dx.doi.org/10.1021/acs.macromol.8b02594](https://doi.org/10.1021/acs.macromol.8b02594)), and data presented by the authors is also ordinary. It is difficult to expect more interesting characteristics from this polymer. Moreover, unfortunately, the authors didn't seem to find remarkable properties that may attract attention.

Thank you for pointing this out. We are aware that the synthesis of poly(isohexide succinates) has been reported relatively often in the polymer literature (see also Fig. 1 (formerly Extended Data Fig. 1) on all previous synthesis attempts of poly(isosorbide succinate)). This actually underlines the high level of interest in this material, combined with inability to make it at sufficient molecular weight, as these previous efforts were never able to produce the polymer in question with a sufficiently high molecular weight to even characterize barrier or mechanical properties, let alone for any interesting application (even when starting the synthesis from highly reactive diacid chlorides, as recently reported by Liu *et al.* <https://doi.org/10.1016/j.eurpolymj.2020.109846> on the synthesis poly(isosorbide succinate) and poly(isomannide succinate)).

The publication by Marubayashi *et al.* pointed out by the reviewer is referred to in Table 3 (formerly Table 1), entry poly(isomannide succinate) (Ref. [32]) of our manuscript). The authors obtained the material by polymerization of succinyl chloride, which is highly reactive and requires the use of stoichiometric amounts of reactive compounds such as SOCl_2 . This synthesis route has several drawbacks. It is very expensive because it requires stoichiometric reagents (very unsustainable as well) and corrosion-resistant reactors (due to the release of HCl during the polymerization) as well as purification of the polymer product by precipitation from an organic solvent, which also decreases the yield of the polymer product. Our goal in the present manuscript was to develop a less wasteful synthesis method that enables the synthesis of these fully isohexide-based polyesters. Incidentally, the present synthesis method is superior to the established synthesis from diacid chlorides not only regarding the amount of reaction steps and waste created. It also yields polyesters with a higher molecular weight than typically obtained reactions starting from diacid chlorides (as seen in some examples in Table 3 (formerly Table 1)).

Regarding the properties of the materials described in our manuscript: To our knowledge, the mechanical and barrier properties of the respective materials have been reported for the first time in the literature. The comparisons we draw to established materials show that materials like poly(isosorbide succinate) have indeed some unique properties compared to established, fossil-based materials. The present manuscript focuses on this new type of synthesis method. With high molecular weight material in hand, we can now explore possible applications of these polymers.

- The authors didn't provide mechanical data of 3D printed objects.

Indeed a good point, we did not characterize these properties for 3D printed objects. Another reviewer for this manuscript made a similar remark. We are not planning to include further characterization of the mechanical properties of 3D printed objects in the current manuscript, as we believe that it is out of scope considering the focus of the paper on the new synthesis method (see also reply to Reviewer 1).

- In the section of Chemical recycling and end of life, data is insufficient to discuss the meaning for sustainability.

Concerning chemical recycling: A valid point, as no chemical recycling experiments have been discussed. We toned down the focus on chemical recycling in the manuscript and instead referred more to bio- and hydrolytic degradability, where experiments have been performed.

Concerning the biodegradability, we are currently investigating ways to induce biodegradability of the described polymers by chemical modification, although those results are not yet ready to be published. We believe nonetheless that the preliminary results on the biodegradability and hydrolytic degradability of poly(isosorbide succinate) show that the degradability seems to be influenced by the high molecular weights we obtained in our study (as opposed to previous results by Okada *et al.* on the same polymer with significantly lower molecular weights of $M_n \sim 7.5$ kg/mol, see Refs [12, 31]). These results are preliminary owing to the long timespans required to properly investigate the degradability of polymers under ambient conditions. We expect that poly(isosorbide succinate) should exhibit some form of degradability due to the previously reported results by Okada *et al.*

Minor points:

-No proper abstract was provided.

The initial submission of this paper was aimed at another journal with different abstract guidelines. We adjusted our abstract to Nature Communication guidelines in the revised submission.

-In Fig. 1, illustrated isosorbide repeating unit in the polymer chain must lose the chirality. It should be revised.

While this might be true from a strictly chemical point of view, we believe retaining the chirality of isosorbide in the polymer chain is important for clarity. This is because we also report isomannide-based polyesters in our manuscript, which could create confusion if chirality is lost. A look in the literature shows that many authors choose to retain isosorbide's chirality in polymer chains.

Reviewer #3 (Remarks to the Author):

The manuscript by Weinland *et al.* is an interesting report of improving molecular weight and thus properties in isosorbide monomers via step growth polymerization. Overall, the data is compelling and important to the field of sustainable polymers and this reviewer enjoyed reading the article and supportive of publication with revisions. There are a few things below that should be addressed prior to publication:

1) What is the mole ratio of p-cresol?

Reactions were typically conducted with 1.5 equivalents of p-cresol relative to the diacid moiety, which was mentioned both in the main text (subheading 'Effect of aryl alcohols during esterification and polycondensation') and the Methods section. Future work will focus on reducing that amount to fully utilize reactor volumes.

2) Do the kinetics follow Carother's equation?

We did observe a strong dependency of the final molecular weight of the polymer on an equimolar ratio of diacid and diol reactant. If even a slight excess of diol is added the molecular weights of the polymer product are significantly decreased. This observation is in accordance with Carother's equation, where an equimolar reactant ratio (this includes high

monomer purity) and a high monomer conversion (typically $\geq 99\%$) are required to obtain high degrees of polymerization. It is likely related to the high boiling point ($\sim 372\text{ }^\circ\text{C}$) of isosorbide and thus the challenging removal of its excess from the polymer melt. We included a short discussion of the importance of this equimolar ratio of diol and diacid in the revised manuscript.

3) What is the typical T_g s of the lower molecular weight isosorbide esters (it would be good to report these for comparison)?

Indeed an important comparison. We provided the T_g and M_n values of previously published poly(isosorbide succinate) in Fig.1 (formerly Extended Data Fig.1). T_g values tend to be lower (as low as $36\text{ }^\circ\text{C}$) compared to our reported values of $T_g 82\text{ }^\circ\text{C}$.

4) What is the price point of the final product and how does it compare to current commercial polymers (particularly isosorbide-containing polymers) used in industrial products?

We have done calculations on this within our group. The price of a polymer like PIsSu heavily depends on the reaction scales. If a bulk reaction scale ($>kt$) is assumed, the price of a polymer like PIsSu can compete with existing polymer like PET or commercial isosorbide-containing polyesters like PEIT ($1\text{-}2\text{ }^\text{€}/\text{kg}$). At scales relevant for initial upscaling experiments (upper kg to ton scale) the price of new polyesters like PIsSu is still significantly higher. Unfortunately, we can not share estimated prices in the manuscript because they contain confidential information.

5) Why are the PDIs so high (>2.0), which is beyond Carother's theory for step-growth at 100% conversion? Are there side reactions occurring for these polymers?

As seen in Table 3 (formerly Table 1) of the manuscript, all isosorbide-based polyesters have a PDI < 2.2 , which, in our experience, is a normal range for PDI values of polyesters obtained by melt polymerization. An example of polyesters with high PDI values can be seen in the work of Yoon *et al.* ([dx.doi.org/10.1021/ma4001435](https://doi.org/10.1021/ma4001435), also cited in Table 3, entry on poly(isosorbide-1,4-cyclohexanedicarboxylate)). The authors reported PDI values as high as 5.6.

In the case of poly(isosorbide succinate) we have observed that PDIs can indeed increase by the formation of side products, namely 1,4-sorbitan (ring opening hydration product of isosorbide, see also manuscript part on Scale-up). In the case of isomannide-based polyesters (Table 3), PDIs are relatively high in some cases. Here we did not observe any side product in the ^1H or ^{13}C NMR spectra of the respective products. There could be some side reaction, the product of which overlaps with repeat unit peaks of isomannide-based polyesters in ^1H NMR. It should also be noted that isomannide has not been purified as thoroughly prior to polymerization (only recrystallized from acetone) as isosorbide (recrystallized and distilled, also commercially available with very high purities for polyester synthesis). It should also be noted that literature comparison values in Table 3 (shown in brackets) are often determined after the polymer was precipitated from an organic solvent (also indicated with ^b after the values), which typically reduces PDI values.

6) Generally, the figures need work and quality improved for publication.

We improved the figures in the revised manuscript.

7) Why is the material yellow-brown in color (is it partly degrading during the printing)?

The yellow-brown colouration of the material was observed in all reaction products. The materials themselves should be colourless, the observed colouration is caused by the presence of degradation products. Isosorbide-based polyesters, especially those containing large amounts of isosorbide, have been known in the literature to easily discolour during polymer synthesis. Similarly, we observe the discolouration seen on the provided pictures mainly occurring during polymer synthesis (mainly during esterification it seems). We could not identify any degradation products by NMR, although that is rarely possible due to the minute amounts required to be present in a polymer material to induce discolouration.

Additionally it should be noted that the aryl alcohols used in this study are also known to be easily oxidized, forming coloured by-products (sterically hindered phenols are commonly used as primary antioxidants in polyester synthesis due to this facile oxidative degradation). It could thus also be that some of the colouration occurs by oxidative degradation of aryl alcohols during polyester synthesis.

We hope to address the colouration of the polymer products with further reaction optimization to minimize colour-inducing side reactions. Larger reaction scales and higher monomer purities can also contribute to solving this.

REVIEWERS' COMMENTS

Reviewer #1 (Remarks to the Author):

From the authors response, all points have been addressed in an acceptable manner. Concerning the 3D printing, yes, I would suggest to remove this data if it is not further substantiated.

Reviewer #3 (Remarks to the Author):

The authors have responded to my reviews in a satisfactory manner.

REVIEWERS' COMMENTS

Reviewer #1 (Remarks to the Author):

From the authors response, all points have been addressed in an acceptable manner. Concerning the 3D printing, yes, I would suggest to remove this data if it is not further substantiated.

We have removed the paragraph on 3D printing from the manuscript.

Reviewer #3 (Remarks to the Author):

The authors have responded to my reviews in a satisfactory manner.